# Newton Losses: Using Curvature Information for Learning with Differentiable Algorithms

**Felix Petersen**
Stanford University
`mail@felix-petersen.de`

**Christian Borgelt**
University of Salzburg
`christian@borgelt.net`

**Tobias Sutter**
University of Konstanz
`tobias.sutter@uni.kn`

**Hilde Kuehne**
Tuebingen AI Center
MIT-IBM Watson AI Lab
`h.kuehne@uni-tuebingen.de`

**Oliver Deussen**
University of Konstanz
`oliver.deussen@uni.kn`

**Stefano Ermon**
Stanford University
`ermon@cs.stanford.edu`

## Abstract

When training neural networks with custom objectives, such as ranking losses and shortest-path losses, a common problem is that they are, per se, non-differentiable. A popular approach is to continuously relax the objectives to provide gradients, enabling learning. However, such differentiable relaxations are often non-convex and can exhibit vanishing and exploding gradients, making them (already in isolation) hard to optimize. Here, the loss function poses the bottleneck when training a deep neural network. We present Newton Losses, a method for improving the performance of existing hard to optimize losses by exploiting their second-order information via their empirical Fisher and Hessian matrices. Instead of training the neural network with second-order techniques, we only utilize the loss function's second-order information to replace it by a Newton Loss, while training the network with gradient descent. This makes our method computationally efficient. We apply Newton Losses to eight differentiable algorithms for sorting and shortest-paths, achieving significant improvements for less-optimized differentiable algorithms, and consistent improvements, even for well-optimized differentiable algorithms.

## 1 Introduction

Traditionally, fully-supervised classification and regression learning relies on convex loss functions such as MSE or cross-entropy, which are easy-to-optimize in isolation. However, the need for large amounts of ground truth annotations is a limitation of fully-supervised learning; thus, weakly-supervised learning with non-trivial objectives [1]–[7] has gained popularity. Rather than using fully annotated data, these approaches utilize problem-specific algorithmic knowledge incorporated into the loss function via a continuous relaxation. For example, instead of supervising ground truth values, supervision can be given in the form of ordering information (ranks), e.g., based on human preferences [8], [9]. However, incorporating such knowledge into the loss can make it difficult to optimize, e.g., by making the loss non-convex in the model output, introducing bad local minima, and importantly leading to vanishing as well as exploding gradients, slowing down training [10], [11].

Loss functions that integrate problem-specific knowledge can range from rather simple contrastive losses [12] to rather complex losses that require the integration of differentiable algorithms [2], [7], [8], [11], [13]. In this work, we primarily focus on the (harder) latter category, which allows for solving specialized tasks such as inverse rendering [14]–[16], learning-to-rank [2], [5], [8], [11], [17]–[20], self-supervised learning [3], differentiation of optimizers [21], [22], and top-k supervision [2], [5], [23]. In this paper, we summarize these loss functions under the umbrella of algorithmic losses [24] as they introduce algorithmic knowledge via continuous relaxations into the training objective.

While the success of neural network training is primarily due to the backpropagation algorithm and stochastic gradient descent (SGD), there is also a promising line of work on second-order optimization for neural network training [25]–[33]. Compared to first-order methods like SGD, second-order optimization methods exhibit improved convergence rates and therefore require fewer training steps; however, they have two major limitations [31], namely (i) computing the inverse of the curvature matrix for a large and deep neural network is computationally substantially more expensive than simply computing the gradient with backpropagation, which makes second-order methods practically inapplicable in many cases [34]; (ii) networks trained with second-order information have been shown to exhibit reduced generalization capabilities [35].

Inspired by ideas from second-order optimization, in this work, we propose a novel method for incorporating second-order information into training with non-convex and hard to optimize algorithmic losses. Loss functions are usually cheaper to evaluate than a neural network. Further, loss functions operate on lower dimensional spaces than those spanned by the parameters of neural networks. If the loss function becomes the bottleneck in the optimization process because it is difficult to optimize, it suggests to use a stronger optimization method that requires fewer steps like second-order optimization. However, as applying second-order methods to neural networks is expensive and limits generalization, we want to train the neural network with first-order SGD. Therefore, we propose Newton Losses, a method for locally approximating loss functions with a quadratic with second-order Taylor expansion. Thereby, Newton Losses provides a (locally) convex loss leading to better optimization behavior, while training the actual neural network with gradient descent.

For the quadratic approximation of the algorithmic losses, we propose two variants of Newton Losses: (i) *Hessian-based Newton Losses*, which comprises a generally stronger method but requires an estimate of the Hessian [31]. Depending on the choice of differentiable algorithm, choice of relaxation, or its implementation, the Hessian may, however, not be available. Thus, we further relax the method to (ii) *empirical Fisher matrix-based Newton Losses*, which derive the curvature information from the empirical Fisher matrix [36], which depends only on the gradients. The empirical Fisher variant can be easily implemented on top of existing algorithmic losses because it does not require to compute their second derivatives, while the Hessian variant requires computation of second derivatives and leads to greater improvements when available.

We evaluate Newton Losses for an array of eight families of algorithmic losses on two popular algorithmic benchmarks: the four-digit MNIST sorting benchmark [37] and the Warcraft shortest-path benchmark [22]. We find that Newton Losses leads to consistent performance improvements for each of the algorithms—for some of the algorithms (those which suffer the most from vanishing and exploding gradients) more than doubling the accuracy.

## 2   Background & Related Work

The related work comprises algorithmic supervision losses and second-order optimization methods. To the best of our knowledge, this is the first work combining second-order optimization of loss functions with first-order optimization of neural networks, especially for algorithmic losses.

**Algorithmic Losses.**   Algorithmic losses, i.e., losses that contain some kind of algorithmic component, have become quite popular in recent machine learning research. In the domain of recommender systems, early learning-to-rank works already appeared in the 2000s [17], [18], [38], but more recently Lee *et al.* [39] proposed differentiable ranking metrics, and Swezey *et al.* [8] proposed PiRank, which relies on differentiable sorting. For differentiable sorting, an array of methods has been proposed in recent years, which includes NeuralSort [37], SoftSort [40], Optimal Transport Sort [2], differentiable sorting networks (DSN) [5], and the relaxed Bubble Sort algorithm [24]. Other works explore differentiable sorting-based top-k for applications such as differentiable image patch selection [41], differentiable k-nearest-neighbor [23], [37], top-k attention for machine translation [23], differentiable beam search methods [23], [42], survival analysis [43], and self-supervised learning [7]. But algorithmic losses are not limited to sorting: other works have considered learning shortest-paths [21], [22], [24], [44], learning 3D shapes from images and silhouettes [14]–[16], [24], [45]–[47], learning with combinatorial solvers for NP-hard problems [22], learning to classify handwritten characters based on editing distances between strings [24], learning with differentiable physics simulations [48], and learning protein structure with a differentiable simulator [49], among many others.

**Second-Order Optimization.** Second-order methods have gained popularity in machine learning due to their fast convergence properties when compared to first-order methods [25]. One alternative to the vanilla Newton's method are quasi-Newton methods, which, instead of computing an inverse Hessian in the Newton step (which is expensive), approximate this curvature from the change in gradients [31], [50], [51]. In addition, a number of new approximations to the pre-conditioning matrix have been proposed in the literature, i.a., [26], [28], [52]–[54]. While the vanilla Newton method relies on the Hessian, there are variants which use the empirical Fisher matrix, which can coincide in specific cases with the Hessian, but generally exhibits somewhat different behavior. For an overview and discussion of Fisher-based methods (including natural gradient descent), see [36], [55].

## 3 Newton Losses

### 3.1 Preliminaries

We consider the training of a neural network $f(x; \theta)$, where $x \in \mathbb{R}^n$ is the vector of inputs, $\theta \in \mathbb{R}^d$ is the vector of trainable parameters and $y = f(x; \theta) \in \mathbb{R}^m$ is the vector of outputs. As per vectorization, $\mathbf{x} = [x_1, \ldots, x_N]^\top \in \mathbb{R}^{N \times n}$ denotes a set of $N$ input data points, and $\mathbf{y} = f(\mathbf{x}; \theta) \in \mathbb{R}^{N \times m}$ denotes the neural network outputs corresponding to the inputs. Further, let $\ell : \mathbb{R}^{N \times m} \to \mathbb{R}$ denote the loss function, and let the "label" information be implicitly encoded in $\ell$. The reason for this choice of implicit notation is that, for many algorithmic losses, it is not just a label, e.g., it can be ordinal information between multiple data points or a set of encoded constraints. We assume the loss function to be twice differentiable, but also present an extension for only once differentiable losses, as well as non-differentiable losses via stochastic smoothing in the remainder of the paper.

Conventionally, the parameters $\theta$ are optimized using an iterative algorithm (e.g., SGD [56], Adam [57], or Newton's method [31]) that updates them repeatedly according to:

$$\theta_t \leftarrow \text{One optim. step of } \ell(f(\mathbf{x}; \theta)) \text{ wrt. } \theta \text{ at } \theta = \theta_{t-1} . \tag{1}$$

However, in this work, we consider splitting this optimization update step into two alternating steps:

$$\mathbf{z}_t^\star \leftarrow \text{One optim. step of } \ell(\mathbf{z}) \text{ wrt. } \mathbf{z} \text{ at } \mathbf{z} = f(\mathbf{x}; \theta_{t-1}) , \tag{2a}$$

$$\theta_t \leftarrow \text{One optim. step of } \tfrac{1}{2}\|\mathbf{z}_t^\star - f(\mathbf{x}; \theta)\|_2^2 \text{ wrt. } \theta \text{ at } \theta = \theta_{t-1} . \tag{2b}$$

More formally, this can also be expressed via a function $\phi(\,\cdot\,, \cdot\,, \cdot\,)$ that describes one update step (its first argument is the objective to be minimized, its second argument is the variable to be optimized, and its third argument is the starting value for the variable) as follows:

$$\theta_t \leftarrow \phi(\,\ell(f(\mathbf{x}; \theta)), \qquad \theta, \qquad \theta_{t-1}\,) \tag{3}$$

And, for two update step functions $\phi_1$ and $\phi_2$, we can formalize (2) to

$$\mathbf{z}_t^\star \leftarrow \phi_1(\,\ell(\mathbf{z}), \qquad \mathbf{z}, \qquad f(\mathbf{x}; \theta_{t-1})\,) , \tag{4a}$$

$$\theta_t \leftarrow \phi_2(\,\tfrac{1}{2}\|\mathbf{z}_t^\star - f(\mathbf{x}; \theta)\|_2^2, \; \theta, \; \theta_{t-1}\,) . \tag{4b}$$

The purpose of the split is to enable us to use two different iterative optimization algorithms $\phi_1$ and $\phi_2$. This is particularly interesting for optimization problems where the optimization of the loss function $\ell$ is a difficult optimization problem. For standard convex losses like MSE or CE, gradient descent is a perfectly sufficient choice for $\phi_1$ (MSE will recover the goal, and CE leads to outputs $\mathbf{z}^\star$ that achieve a perfect argmax classification result). However, if there is the asymmetry of $\ell$ being harder to optimize (requiring more steps), while (4a) being much cheaper per step compared to (4b), then the optimization of the loss (4a) comprises a bottleneck compared to the optimization of the neural network (4b). Such conditions are prevalent in the space of algorithmic supervision losses.

A similar split (for the case of splitting between the layers of a neural network, and using gradient descent for both (2a) and (2b), i.e., the requirement of $\phi_1 = \phi_2$) is also utilized in the fields of biologically plausible backpropagation [58]–[61] and proximal backpropagation [62], leading to reparameterizations of backpropagation. For SGD, we show that (3) is exactly equivalent to (4) in Lemma 2, and for a special case of Newton's method, we show the equivalence in Lemma 3 in the SM. Motivated by the equivalences under the split, in the following, we consider the case of $\phi_1 \neq \phi_2$.

## 3.2 Method

Equipped with the two-step optimization (2) / (4), we can introduce the idea behind Newton Losses:

*We propose $\phi_1$ to be Newton's method, while $\phi_2$ remains stochastic gradient descent.*

In the following, we formulate how we can solve optimizing (2a) with Newton's method, or, whenever we do not have access to the Hessian of $\ell$, using a step pre-conditioned via the empirical Fisher matrix. This allows us to transform an original loss function $\ell$ into a Newton loss $\ell^*$, which allows optimizing $\ell^*$ with gradient descent only while maintaining equivalence to the two-step idea, and thereby making it suitable for common machine learning frameworks.

Newton's method relies on a quadratic approximation of the loss function at location $\bar{\mathbf{y}} = f(\mathbf{x}; \theta)$

$$\tilde{\ell}_{\bar{\mathbf{y}}}(\mathbf{z}) = \ell(\bar{\mathbf{y}}) + (\mathbf{z} - \bar{\mathbf{y}})^\top \nabla_{\bar{\mathbf{y}}} \ell(\bar{\mathbf{y}}) + \tfrac{1}{2}(\mathbf{z} - \bar{\mathbf{y}})^\top \nabla^2_{\bar{\mathbf{y}}} \ell(\bar{\mathbf{y}}) (\mathbf{z} - \bar{\mathbf{y}}),  \tag{5}$$

and sets its derivative to 0 to find the location $\mathbf{z}^\star$ of the stationary point of $\tilde{\ell}_{\bar{\mathbf{y}}}(\mathbf{z})$:

$$\nabla_{\mathbf{z}^\star} \tilde{\ell}_{\bar{\mathbf{y}}}(\mathbf{z}^\star) = 0 \;\; \Leftrightarrow \;\; \nabla_{\bar{\mathbf{y}}} \ell(\bar{\mathbf{y}}) + \nabla^2_{\bar{\mathbf{y}}} \ell(\bar{\mathbf{y}})(\mathbf{z}^\star - \bar{\mathbf{y}}) = 0 \;\; \Leftrightarrow \;\; \mathbf{z}^\star = \bar{\mathbf{y}} - (\nabla^2_{\bar{\mathbf{y}}} \ell(\bar{\mathbf{y}}))^{-1} \nabla_{\mathbf{y}} \ell(\bar{\mathbf{y}}). \tag{6}$$

However, when $\ell$ is non-convex or the smallest eigenvalues of $\nabla^2_{\bar{\mathbf{y}}} \ell(\bar{\mathbf{y}})$ either become negative or zero, this $\mathbf{z}^\star$ may not be a good proxy for a minimum of $\ell$, but may instead be any other stationary point or lie far away from $\bar{\mathbf{y}}$, leading to exploding gradients downstream. To resolve this issue, we introduce Tikhonov regularization [63] with a strength of $\lambda$, which leads to a well-conditioned curvature matrix:

$$\mathbf{z}^\star = \bar{\mathbf{y}} - (\nabla^2_{\bar{\mathbf{y}}} \ell(\bar{\mathbf{y}}) + \lambda \cdot \mathbf{I})^{-1} \nabla_{\bar{\mathbf{y}}} \ell(\bar{\mathbf{y}}). \tag{7}$$

Using $\mathbf{z}^\star$, we can plug the solution into (2b) to find the Newton loss $\ell^*$ and compute its derivative as

$$\ell^*_{\mathbf{z}^\star}(\mathbf{y}) = \tfrac{1}{2}(\mathbf{z}^\star - \mathbf{y})^\top (\mathbf{z}^\star - \mathbf{y}) = \tfrac{1}{2} \| \mathbf{z}^\star - \mathbf{y} \|_2^2 \qquad \text{and} \qquad \nabla_{\mathbf{y}} \ell^*_{\mathbf{z}^\star}(\mathbf{y}) = \mathbf{y} - \mathbf{z}^\star. \tag{8}$$

Here, as in Section 3.1, $\mathbf{y} = f(\mathbf{x}, \theta)$. Via this construction, we obtain the Newton loss $\ell^*_{\mathbf{z}^\star}$, a new convex loss, which itself has a gradient that corresponds to one Newton step of the original loss. In particular, on $\mathbf{y}$, one gradient descent step on the Newton loss (8) reduces to

$$\mathbf{y} \;\; \leftarrow \;\; \mathbf{y} - \eta \cdot \nabla_{\mathbf{y}} \ell^*_{\mathbf{z}^\star}(\mathbf{y}) = \mathbf{y} - \eta \cdot (\mathbf{y} - \mathbf{z}^\star) \;\; = \;\; \mathbf{y} - \eta \cdot (\nabla^2_{\mathbf{y}} \ell(\mathbf{y}) + \lambda \cdot \mathbf{I})^{-1} \nabla_{\mathbf{y}} \ell(\mathbf{y}), \tag{9}$$

which is exactly one step of Newton's method on $\mathbf{y}$. Thus, we can optimize the Newton loss $\ell^*_{\mathbf{z}^\star}(f(\mathbf{x}; \theta))$ with gradient descent, and obtain equivalence to the proposed concept.

In the following definition, we summarize the resulting equations that define the Newton loss $\ell^*_{\mathbf{z}^\star}$.

**Definition 1** (Newton Losses (Hessian)). *For a loss function $\ell$ and a given current parameter vector $\theta$, we define the Hessian-based Newton loss via the empirical Hessian as*

$$\ell^*_{\mathbf{z}^\star}(\mathbf{y}) = \tfrac{1}{2} \| \mathbf{z}^\star - \mathbf{y} \|_2^2 \qquad \text{where} \qquad z_i^\star = \bar{y}_i - \left( \tfrac{1}{N} \sum_{j=1}^N \nabla^2_{\bar{y}_j} \ell(\bar{\mathbf{y}}) + \lambda \mathbf{I} \right)^{-1} \nabla_{\bar{y}_i} \ell(\bar{\mathbf{y}}) \tag{10}$$

*for all $i \in \{1, ..., N\}$ and $\bar{\mathbf{y}} = f(\mathbf{x}; \theta)$.*

We remark that computing and inverting the Hessian of the loss function is usually computationally efficient. (We remind the reader that the Hessian of the loss function is the second derivative wrt. the inputs of the loss function and we further remind that the inputs to the loss are **not** the neural network parameters / weights.) Whenever the Hessian matrix of the loss function is not available, whether it may be due to limitations of a differentiable algorithm, large computational cost, lack of a respective implementation of the second derivative, etc., we may resort to using the empirical Fisher matrix (i.e., the second uncentered moments of the gradients) as a source for curvature information. We remark that the empirical Fisher matrix is not the same as the Fisher information matrix [36], and that the Fisher information matrix is generally not available for algorithmic losses. While the empirical Fisher matrix, as a source for curvature information, may be of lower quality than the Hessian matrix, it has the advantage that it can be computed from the gradients, i.e.,

$$\mathbf{F} = \mathbb{E}_x \left[ \nabla_{f(x,\theta)} \ell(f(x,\theta)) \cdot \nabla_{f(x,\theta)} \ell(f(x,\theta))^\top \right]. \tag{11}$$

This means that, assuming a moderate dimension of the prediction space $m$, computing the empirical Fisher comes at no significant overhead and may, conveniently, be performed in-place as we discuss later. Again, we regularize the matrix via Tikhonov regularization with strength $\lambda$ and can, accordingly, define the empirical Fisher-based Newton loss as follows.

| **Algorithm 1** Training with a Newton Loss | **Algorithm 2** Training with InjectFisher |
|---|---|

```python
# Python style pseudo-code
model = ...        # neural network
loss = ...         # original loss fn
optimizer = ...    # optim. of model
tik_l = ...        # hyperparameter

for data, label in data_loader:
  # apply a neural network model
  y = model(data)

  # compute gradient of orig. loss
  grad = gradient(loss(y, label), y)
  # compute Hessian (or alt. Fisher)
  hess = hessian(loss(y, label), y)

  # compute the projected optimum
  z_star = (y - grad @ inverse(hess
    + tik_l * eye(g.shape[1]))).detach()

  # compute the Newton loss
  l = MSELoss()(y, z_star)

  # backpropagate and optim. step
  l.backward()
  optimizer.step()
```

```python
# implements the Fisher-based Newton
# loss via an injected modification
# of the backward pass:
class InjectFisher(AutoGradFunction):
  def forward(ctx, x, tik_l):
    assert len(x.shape) == 2
    ctx.tik_l = tik_l
    return x

  def backward(ctx, g):
    fisher = g.T @ g * g.shape[0]
    input_grad = g @ inverse(fisher
        + ctx.tik_l * eye(g.shape[1]))
    return input_grad, None

for data, label in data_loader:
  # apply a neural network model
  y = model(data)
  # inject the Fisher backward mod.
  y = InjectFisher.apply(y, tik_l)
  # compute the original loss
  l = loss(y, label)
  # backpropagate and optim. step
  l.backward()
  optimizer.step()
```

**Definition 2** (Newton Loss (Fisher)). *For a loss function $\ell$, and a given current parameter vector $\theta$, we define the empirical Fisher-based Newton loss as*

$$\ell^*_{\mathbf{z}^\star}(\mathbf{y}) = \tfrac{1}{2}\|\mathbf{z}^\star - \mathbf{y}\|_2^2 \qquad where \qquad z_i^\star = \bar{y}_i - \left(\tfrac{1}{N}\sum_{j=1}^{N}\nabla_{\bar{y}_j}\ell(\bar{\mathbf{y}})\,\nabla_{\bar{y}_j}\ell(\bar{\mathbf{y}})^\top + \lambda\mathbf{I}\right)^{-1}\nabla_{\bar{y}_i}\ell(\bar{\mathbf{y}})$$

*for all $i \in \{1, ..., N\}$ and $\bar{\mathbf{y}} = f(\mathbf{x}; \theta)$.*

Before continuing with the implementation, integration, and further computational considerations, we can make an interesting observation. In the case of using the trivial MSE loss, i.e., $\ell(y) = \tfrac{1}{2}\|y - y^\star\|_2^2$ where $y^\star$ denotes a ground truth, the Newton loss collapses to the original MSE loss. This illustrates that Newton Losses requires non-trivial original losses. Another interesting aspect is the arising fixpoint—the Newton loss of a Newton loss is equivalent to a simple Newton loss.

### 3.3 Implementation

After introducing Newton Losses, in this section, we discuss aspects of implementation and illustrate its implementations in Algorithms 1 and 2. Whenever we have access to the Hessian matrix of the algorithmic loss function, it is generally favorable to utilize the Hessian-based approach (Algo. 1 / Def. 1), whereas we can utilize the empirical Fisher-based approach (Algo. 2 / Def. 2) in any case.

In Algorithm 1, the difference to regular training is that we use the original `loss` only for the computation of the gradient (`grad`) and the Hessian matrix (`hess`) of the original loss. Then we compute $\mathbf{z}^\star$ (`z_star`). Here, depending on the automatic differentiation framework, we need to ensure not to backpropagate through the target `z_star`, which may be achieved, e.g., via ".detach()" or ".stop_gradient()", depending on the choice of library. Finally, the Newton loss `l` may be computed as the squared / MSE loss between the model output `y` and `z_star` and an optimization step on `l` may be performed. We note that, while we use a `label` from our `data_loader`, this label may be empty or an abstract piece of information for the differentiable algorithm; in our experiments, we use ordinal relationships between data points as well as shortest-paths on graphs.

In Algorithm 2, we show how to apply the empirical Fisher-based Newton Losses. In particular, due to the empirical Fisher matrix depending only on the gradient, we can compute it in-place during the backward pass / backpropagation, which makes this variant particularly simple and efficient to apply. This can be achieved via an injection of a custom gradient right before applying the original `loss`,

which replaces the gradient in-place by a gradient that corresponds to Definition 2. The injection is performed by the `InjectFisher` function, which corresponds to an identity during the forward pass but replaces the gradient by the gradient of the respective empirical Fisher-based Newton loss.

In both cases, the only additional hyperparameter to specify is the Tikonov regularization strength $\lambda$ (`tik_l`). $\lambda$ heavily depends on the algorithmic loss function, particularly, on the magnitude of gradients provided by the algorithmic loss, which may vary drastically between different methods and implementations. Other factors may be the choice of Hessian / Fisher, the dimension of outputs $m$, the batch size $N$. Notably, for very large $\lambda$, the direction of the gradient becomes more similar to regular gradient descent, and for smaller $\lambda$, the effect of Newton Losses increases. We provide an ablation study for $\lambda$ in Section 4.3.

# 4 Experiments[1]

For the experiments, we apply Newton Losses to eight methods for differentiable algorithms and evaluate them on two established benchmarks for algorithmic supervision, i.e., problems where an algorithm is applied to the predictions of a model and only the outputs of the algorithm are supervised. Specifically, we focus on the tasks of ranking supervision and shortest-path supervision because they each have a range of established methods for evaluating our approach. In ranking supervision, only the relative order of a set of samples is known, while their absolute values remain unsupervised. The established benchmark for differentiable sorting and ranking algorithms is the multi-digit MNIST sorting benchmark [2], [5], [11], [37], [40]. In shortest-path supervision, only the shortest-path of a graph is supervised, while the underlying cost matrix remains unsupervised. The established benchmark for differentiable shortest-path algorithms is the Warcraft shortest-path benchmark [21], [22], [24]. As these tasks require backpropagating through conventionally non-differentiable algorithms, the respective approaches make the ranking or shortest-path algorithms differentiable such that they can be used as part of the loss.

## 4.1 Ranking Supervision

In this section, we explore ranking supervision [37] with an array of differentiable sorting-based losses. Here, we use the four-digit MNIST sorting benchmark [37], where sets of $n$ four-digit MNIST images are given, and the supervision is the relative order of these images corresponding to the displayed value, while the absolute values remain unsupervised. The goal is to learn a CNN that maps each image to a scalar value in an order preserving fashion. As losses, we use sorting supervision losses based on the NeuralSort [37], the SoftSort [40], the logistic Differentiable Sorting Network [5], and the monotonic Cauchy DSN [11]. *NeuralSort* and *SoftSort* work by mapping an input list (or vector) of values to a differentiable permutation matrix that is row-stochastic and indicates the order / ranking of the inputs. *Differentiable Sorting Networks* offer an alternative to NeuralSort and SoftSort. DSNs are based on sorting networks, a classic family of sorting algorithms that operate by conditionally swapping elements. By introducing perturbations, DSNs relax the conditional swap operator to a differentiable conditional swap and thereby continuously relax the sorting and ranking operators. We discuss the background of each of these diff. sorting and ranking algorithms in greater detail in Supplementary Material B.

**Setups.** The sorting supervision losses are cross-entropy losses defined between the differentiable permutation matrix produced by a respective differentiable sorting operator and the ground truth permutation matrix corresponding to a ground truth ranking. The Cauchy DSN may be an exception to the hard to optimize classification as it is quasi-convex [11]. We evaluate the sorting benchmark for numbers of elements to be ranked $n \in \{5, 10\}$ and use the percentage of rankings correctly identified

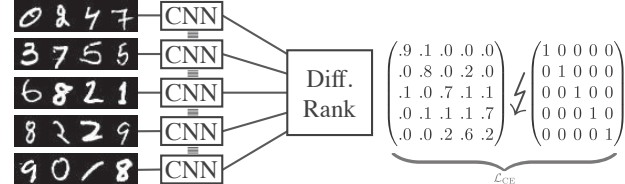

Figure 1: Overview over ranking supervision with a differentiable sorting / ranking algorithm. A set of input images is (element-wise) processed by a CNN, producing a scalar for each image. The scalars are sorted / ranked by the differentiable ranking algorithm, which returns the differentiable permutation matrix, which is compared to the ground truth permutation matrix.

---

[1]Our implementation is openly available at github.com/Felix-Petersen/newton-losses.

Table 1: Ranking supervision with differentiable sorting. The metric is the percentage of rankings correctly identified (and individual element ranks correctly identified, in parentheses) avg. over 10 seeds. Statistically significant improvements (sig. level 0.05) are indicated bold black; improved means are indicated in bold grey.

| $n = 5$ | NeuralSort [37] | | SoftSort [40] | | Logistic DSN [5] | | Cauchy DSN [11] | |
|---|---|---|---|---|---|---|---|---|
| Baseline | 71.33±2.05 | (87.10±0.96) | 70.70±2.60 | (86.75±1.26) | 53.56±18.04 | (77.04±10.30) | 85.09±0.77 | (93.31±0.39) |
| NL (Hessian) | **83.31±1.70** | (**92.54±0.73**) | **83.87±0.81** | (**92.72±0.39**) | **75.02±12.59** | (**88.53±06.00**) | 85.11±0.78 | (93.31±0.34) |
| NL (Fisher) | **83.93±0.62** | (**92.80±0.30**) | **84.03±0.59** | (**92.82±0.24**) | 63.11±30.63 | (79.28±22.16) | 84.95±0.79 | (93.25±0.37) |

| $n = 10$ | NeuralSort | | SoftSort | | Logistic DSN | | Cauchy DSN | |
|---|---|---|---|---|---|---|---|---|
| Baseline | 24.26±01.52 | (74.47±0.83) | 27.46±3.58 | (76.02±1.92) | 12.31±10.22 | (58.81±16.79) | 55.29±2.46 | (87.06±0.85) |
| NL (Hessian) | **48.76±05.88** | (**84.83±2.13**) | **55.07±1.08** | (**86.89±0.31**) | **42.14±22.30** | (**75.35±23.77**) | **56.49±1.02** | (**87.44±0.40**) |
| NL (Fisher) | **39.23±11.38** | (**81.14±4.91**) | **54.00±2.24** | (**86.56±0.68**) | 25.72±27.42 | (52.18±36.51) | **56.12±1.86** | (**87.35±0.65**) |

as well as percentage of individual element ranks correctly identified as evaluation metrics. For each of the four original baseline methods, we compare it to two variants of their Newton losses: the empirical Hessian and the empirical Fisher variant. For each setting, we train the CNN on 10 seeds using the Adam optimizer [57] at a learning rate of $10^{-3}$ for $10^5$ steps and batch size of 100.

**Results.** As displayed in Table 1, we can see that—for each original loss—Newton Losses improve over their baselines. For NeuralSort, SoftSort, and Logistic DSNs, we find that using the Newton losses substantially improves performance. Here, the reason is that these methods suffer from vanishing and exploding gradients, especially for the more challenging case of $n = 10$. As expected, we find that the Hessian Newton Loss leads to better results than the Fisher variant, except for NeuralSort and SoftSort in the easy setting of $n = 5$, where the results are nevertheless quite close. Monotonic differentiable sorting networks, i.e., the Cauchy DSNs, provide an improved variant of DSNs, which have the property of quasi-convexity and have been shown to exhibit much better training behavior out-of-the-box, which makes it very hard to improve upon the existing results. Nevertheless, Hessian Newton Losses are on-par for the easy case of $n = 5$ and, notably, improve the performance by more than $1\%$ on the more challenging case of $n = 10$. To explore this further, we additionally evaluate the Cauchy DSN for $n = 15$ (not displayed in the table): here, the baseline achieves 30.84±2.74 (82.30±1.08), whereas, using NL (Fisher), we improve it to 32.30±1.22 (82.78±0.53), showing that the trend of increasing improvements with more challenging settings (compared to smaller $n$) continues. Summarizing, we obtain strong improvements on losses that are hard to optimize, while in already well-behaving cases the improvements are smaller. This perfectly aligns with our goal of improving performance on losses that are hard to optimize.

## 4.2 Shortest-Path Supervision

In this section, we apply Newton Losses to the shortest-path supervision task of the $12 \times 12$ Warcraft shortest-path benchmark [21], [22], [24]. Here, $12 \times 12$ Warcraft terrain maps are given as $96 \times 96$ RGB images (e.g., Figure 2 left) and the supervision is the shortest path from the top left to the bottom right (Figure 2 right) according to a hidden cost embedding (Figure 2 center).

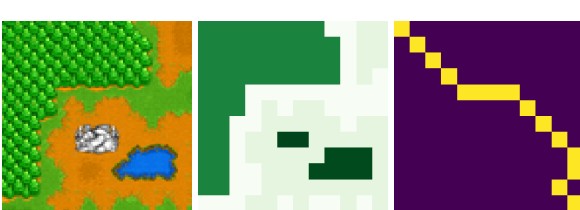

Figure 2: $12 \times 12$ Warcraft shortest-path problem. An input terrain map (left), unsupervised ground truth cost embedding (center) and ground truth supervised shortest path (right).

The hidden cost embedding is not available for training. The goal is to predict $12 \times 12$ cost embeddings of the terrain maps such that the shortest path according to the predicted embedding corresponds to the ground truth shortest path. Vlastelica et al. [22] have shown that integrating an algorithm in the training pipeline substantially improves performance compared to only using a neural network with an easy-to-optimize loss function, which has been confirmed by subsequent work [21], [24]. For this task, we explore a set of families of algorithmic supervision approaches: *Relaxed Bellman-Ford* [24] is a shortest-path algorithm relaxed via the AlgoVision framework, which continuously relaxes algorithms by perturbing all accessed variables with logistic distributions and approximating the expectation value in closed form. *Stochastic Smoothing* [64] is a sampling-based differentiation method that can be used to relax, e.g., a shortest-path algorithm by perturbing the input with probability distribution. *Perturbed Optimizers with Fenchel-Young Losses* [21] build on stochastic smoothing and Fenchel-Young losses [65] and identify the argmax to be the differential of max, which allows a simplification of stochastic smoothing, again applied, e.g., to shortest-path

learning problems. We use the same hyperparameters as shared by previous works [21], [24]. In particular, it is notable that, throughout the literature, the benchmark assumes a training duration of 50 epochs and a learning rate decay by a factor of 10 after 30 and 40 epochs each. Thus, we do not deviate from these constraints.

### 4.2.1 Relaxed Bellman-Ford

The relaxed Bellman-Ford algorithm [24] is a continuous relaxation of the Bellman-Ford algorithm via the AlgoVision library. To increase the number of settings considered, we explore four sub-variants of the algorithm: For+$L_1$, For+$L_2^2$, While+$L_1$, and While+$L_2^2$. Here, For / While refers to the distinction between using a While and For loop in Bellman-Ford, while $L_1$ vs. $L_2^2$ refer to the choice of metric between shortest paths. As computing the Hessian of the AlgoVision Bellman-Ford algorithm is too expensive with the PyTorch implementation, for this evaluation, we restrict it to the empirical Fisher-based Newton loss. The results displayed in Table 2. While the differences are rather small, as the baseline here is already strong, we can observe improvements in all of the four settings and in one case achieve a significant improvement. This can be attributed to (i) the high performance of the baseline algorithm on this benchmark, and (ii) that only the empirical Fisher-based Newton loss is available, which is not as strong as the Hessian variant.

Table 2: Shortest-path benchmark results for different variants of the AlgoVision-relaxed Bellman-Ford algorithm [24]. The metric is the percentage of perfect matches averaged over 10 seeds. Significant improvements are bold black, and improved means are bold grey.

| Variant | For+$L_1$ | For+$L_2^2$ | While+$L_1$ | While+$L_2^2$ |
|---|---|---|---|---|
| Baseline | 94.19±0.33 | 95.90±0.21 | 94.30±0.20 | 95.77±0.41 |
| NL (Fisher) | **94.52±0.34** | **96.08±0.46** | **94.47±0.34** | **95.94±0.27** |

### 4.2.2 Stochastic Smoothing

After discussing the analytical relaxation, we continue with stochastic smoothing approaches. First, we consider stochastic smoothing [64], which allows perturbing the input of a function with an exponential family distribution to estimate the gradient of the smoothed function. For a reference on stochastic smoothing with a focus on differentiable algorithms, we refer to the author's recent work [44]. For the baseline, we apply stochastic smoothing to a hard non-differentiable Dijkstra algorithm based loss function to relax it via Gaussian noise ("SS of loss"). We utilize variance reduction via the method of covariates. As we detail in Supplementary Material B.4, stochastic smoothing can also be used to estimate the Hessian of the smoothed function. Based on this result, we can construct the Hessian-variant Newton loss. As an extension to stochastic smoothing, we apply stochastic smoothing only to the non-differentiable Dijkstra algorithm (thereby computing its Jacobian matrix) but use a differentiable loss to compare the predicted relaxed shortest-path to the ground truth shortest-path ("SS of algorithm"). In this case, the Hessian Newton loss is not applicable because the output of the smoothed algorithm is high dimensional and the Hessian of the loss becomes intractable. An extended discussion of the "SS of algorithm" fomulation can be found in SM B.4.1. Nevertheless, we can apply the Fisher-based Newton loss. We evaluate both approaches for 3, 10, and 30 samples.

In Table 3, we can observe that Newton Losses improves the results for stochastic smoothing in each case with more than 3 samples. The reason for the poor performance on 3 samples is that the Hessian or empirical Fisher, respectively, is estimated using only 3 samples, which makes the estimate unstable. For 10 and 30 samples, the performance improves compared to the original method. In Figure 3, we display a respective accuracy plot. When comparing "SS of loss" and "SS of algorithm", we can observe that the extension to smoothing only the algorithm improves performance for at least 10 samples. Here, the reason, again, is that smoothing the algorithm itself requires estimating the Jacobian instead of only the gradient; thus, a larger number of samples is necessary; however starting at 10 samples, smoothing the algorithm performs better, which means that the approach is better at utilizing a given sample budget.

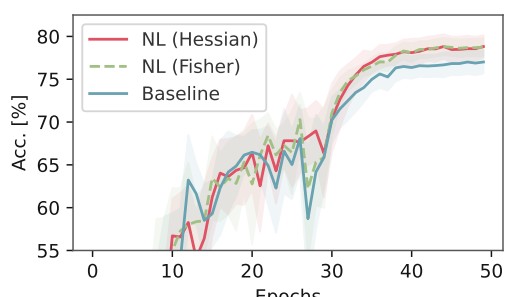

Figure 3: Test accuracy (perfect matches) plot for 'SS of loss' with 10 samples on the Warcraft shortest-path benchmark. Lines show the mean and shaded areas show the 95% conf. intervals.

Table 3: Shortest-path benchmark results for the stochastic smoothing of the loss (including the algorithm), stochastic smoothing of the algorithm (excluding the loss), and perturbed optimizers with the Fenchel-Young loss. The metric is the percentage of perfect matches averaged over 10 seeds. Significant improvements are bold black, and improved means are bold grey.

| Method | SS of loss | | | SS of algorithm | | | PO w/ FY loss | | |
|---|---|---|---|---|---|---|---|---|---|
| # Samples | 3 | 10 | 30 | 3 | 10 | 30 | 3 | 10 | 30 |
| Baseline | **62.83**±5.29 | 77.01±2.18 | 85.48±1.23 | **57.55**±4.58 | 78.70±1.90 | 87.26±1.50 | 80.64±0.75 | 80.39±0.57 | 80.71±1.28 |
| NL (Hessian) | 62.40±5.48 | **78.82**±2.12 | **85.94**±1.33 | — | — | — | **83.09**±3.11 | 81.13±3.58 | **83.45**±2.21 |
| NL (Fisher) | 58.80±5.10 | **78.74**±1.68 | **86.10**±0.60 | 53.82±8.45 | **79.24**±1.78 | 87.41±1.13 | **80.70**±0.65 | 80.37±0.98 | 80.45±0.78 |

### 4.2.3 Perturbed Optimizers with Fenchel-Young Losses

Perturbed optimizers with a Fenchel-Young loss [21] is a formulation of solving the shortest path problem as an $\arg\max$ problem, and differentiating this problem using stochastic smoothing-based perturbations and a Fenchel-Young loss. By extending their formulation to computing the Hessian of the Fenchel-Young loss, we can compute the Newton loss, and find that we can achieve improvements of more than $2\%$. However, for Fenchel-Young losses, which are defined via their derivative, the empirical Fisher is not particularly meaningful, leading to equivalent performance between the baseline and the Fisher Newton loss. Berthet *et al.* [21] mention that their approach works well for small numbers of samples, which we can confirm as seen in Table 3 where the accuracy is similar for each number of samples. An interesting observation is that perturbed optimizers with Fenchel-Young losses perform better than stochastic smoothing in the few-sample regime, whereas stochastic smoothing performs better with larger numbers of samples.

### 4.3 Ablation Study

In this section, we present our ablation study for the (only) hyperparameter $\lambda$. $\lambda$ is the strength of the Tikhonov regularization (see, e.g., Equation 7, or `tik_l` in the algorithms). This parameter is important for controlling the degree to which second-order information is used as well as regularizing the curvature. For the ablation study, we use the experimental setting from Section 4.1 for NeuralSort and SoftSort and $n = 5$. In particular, we consider 13 values for $\lambda$, exploring the range from $0.001$ to $1000$ and plot the element-wise ranking accuracy (individual element ranks correctly identified) in Figure 4. We display the average over 10 seeds as well as each seed's result individually with low opacity. We can observe that Newton Losses are robust over many orders of magnitude for the hyperparameter $\lambda$. Note the logarithmic axis for $\lambda$ in Figure 4. In general, we observe that choices within a few orders of magnitude around $1$ are generally favorable. Further, we observe that NeuralSort is more sensitive to drastic changes in $\lambda$ compared to SoftSort.

### 4.4 Runtime Analysis

We provide tables with runtimes for the experiments in Supplementary Material D. We can observe that the runtimes between the baseline and empirical Fisher-based Newton Losses are indistinguishable for all cases. For the analytical relaxations of differentiable sorting algorithms, where the

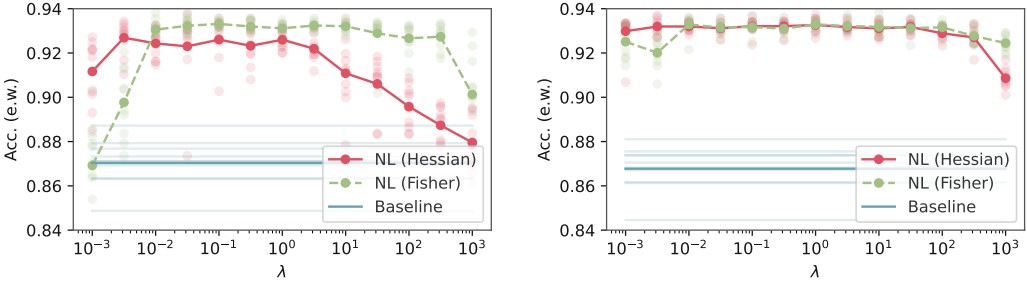

Figure 4: Ablation study wrt. the Tikhonov regularization strength hyperparameter $\lambda$. Displayed is the element-wise ranking accuracy (individual element ranks correctly identified), averaged over 10 seeds, and additionally each seed with low opacity in the background. **Left**: NeuralSort. **Right**: SoftSort. Each for $n = 5$. Newton Losses, and for both the Hessian and the Fisher variant, significantly improve over the baseline for up to (or beyond) 6 orders of magnitude in variation of its hyperparameter $\lambda$. Note the logarithmic horizontal axis.

computation of the Hessian can become expensive with automatic differentiation (i.e., without a custom derivation of the Hessian and without vectorized Hessian computation), we observed overheads between $10\%$ and $2.6\times$. For all stochastic approaches, we observe indistinguishable runtimes for Hessian-based Newton Losses. In summary, applying the Fisher variant of Newton Losses has a minimal computational overhead, whereas, for the Hessian variant, any overhead depends merely on the computation of the Hessian of the algorithmic loss function. While, for differentiable algorithms, the neural network's output dimensionality or algorithm's input dimensionality $m$ is typically moderately small to make the inversion of the Hessian or empirical Fisher cheap, when the output dimensionality $m$ becomes very large such that inversion of the empirical Fisher becomes expensive, we refer to the Woodbury matrix identity [66], which allows simplifying the computation via its low-rank decomposition. A corresponding deviation is included in SM F. Additionally, solver-based inversion implementations can be used to make the inversion more efficient.

## 5    Conclusion

In this work, we focused on weakly-supervised learning problems that require integration of differentiable algorithmic procedures in the loss function. This leads to non-convex loss functions that exhibit vanishing and exploding gradients, making them hard to optimize. We proposed a novel approach for improving performance of algorithmic losses building upon the curvature information of the loss. For this, we split the optimization procedure into two steps: optimizing on the loss itself using Newton's method to mitigate vanishing and exploding gradients, and then optimizing the neural network with gradient descent. We simplified this procedure via a transformation of an original loss function into a Newton loss, which comes in two flavors: a Hessian variant for cases where the Hessian is available and an empirical Fisher variant as an alternative. We evaluated Newton Losses on a set of algorithmic supervision settings, demonstrating that the method can drastically improve performance for weakly-performing differentiable algorithms. We hope that the community adapts Newton Losses for learning with differentiable algorithms and see great potential for combining it with future differentiable algorithms in unexplored territories of the space of differentiable relaxations, algorithms, operators, and simulators.

## Acknowledgments and Disclosure of Funding

This work was in part supported by the IBM-MIT Watson AI Lab, the DFG in the Cluster of Excellence EXC 2117 "Centre for the Advanced Study of Collective Behaviour" (Project-ID 390829875), the Land Salzburg within the WISS 2025 project IDA-Lab (20102-F1901166-KZP and 20204-WISS/225/197-2019), the U.S. DOE Contract No. DE-AC02-76SF00515, the ARO (W911NF-21-1-0125), the ONR (N00014-23-1-2159), and the CZ Biohub.

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

# Appendix

## Table of Contents

## A   Characterization of Meaningful Settings for Newton Losses

For practical purposes, to decide whether applying Newton Losses is expected to improve results, we recommend that a loss $\ell$ fulfills the following 3 minimal criteria:

- (i) non-convex,
- (ii) smoothly differentiable,
- (iii) cannot be solved by a single GD step.

Regarding (ii), the stochastic smoothing formulation enables any non-differentiable function (such as Shortest-paths as considered in this work) to become smoothly differentiable.

Regarding (iii), we note that, e.g., for the MSE loss, the optimum of the loss (when optimizing loss inputs) can be found using a single step of GD (see last paragraph of Section 3.2). For the cross-entropy classification loss, a single GD step leads to the correct class.

## B   Algorithmic Supervision Losses

In this section, we extend the discussion of SoftSort, DiffSort, AlgoVision, and stochastic smoothing.

### B.1   SoftSort and NeuralSort

SoftSort [40] and NeuralSort [37] are prominent yet simple examples of a differentiable algorithm. In the case of ranking supervision, they obtain an array or vector of scalars and return a row-stochastic matrix called the differentiable permutation matrix $P$, which is a relaxation of the argsort operator. Note that, in this case, a set of $k$ inputs yields a scalar for each image and thereby $y \in \mathbb{R}^k$. As a

ground truth label, a ground truth permutation matrix $Q$ is given and the loss between $P$ and $Q$ is the binary cross entropy loss $\ell_{\mathrm{SS}}(y) = \mathrm{BCE}\left(P(y), Q\right)$. Minimizing the loss enforces the order of predictions $y$ to correspond to the true order, which is the training objective. SoftSort is defined as

$$P(y) = \mathrm{softmax}\left(-\left|y^\top \ominus \mathrm{sort}(y)\right|/\tau\right) = \mathrm{softmax}\left(-\left|y^\top \ominus Sy\right|/\tau\right) \tag{12}$$

where $\tau$ is a temperature parameter, "sort" sorts the entries of a vector in non-ascending order, $\ominus$ is the element-wise broadcasting subtraction, $|\cdot|$ is the element-wise absolute value, and "softmax" is the row-wise softmax operator. NeuralSort is defined similarly and omitted for the sake of brevity. In the limit of $\tau \to 0$, SoftSort and NeuralSort converge to the exact ranking permutation matrix [37], [40]. A respective Newton loss can be implemented using automatic differentiation according to Definition 1 or via the empirical Fisher matrix using Definition 2.

## B.2   DiffSort

Differentiable sorting networks (DSN) [5], [11] offer a strong alternative to SoftSort and NeuralSort. They are based on sorting networks, a classic family of sorting algorithms that operate by conditionally swapping elements [67]. As the locations of the conditional swaps are pre-defined, they are suitable for hardware implementations, which also makes them especially suited for continuous relaxation. By perturbing a conditional swap with a distribution and solving for the expectation under this perturbation in closed-form, we can differentiably sort a set of values and obtain a differentiable doubly-stochastic permutation matrix $P$, which can be used via the BCE loss as in Section B.1. We can obtain the respective Newton loss either via the Hessian computed via automatic differentiation or via the Fisher matrix.

## B.3   AlgoVision

AlgoVision [24] is a framework for continuously relaxing arbitrary simple algorithms by perturbing all accessed variables with logistic distributions. The method approximates the expectation value of the output of the algorithm in closed-form and does not require sampling. For shortest-path supervision, we use a relaxation of the Bellman-Ford algorithm [68], [69] and compare the predicted shortest path with the ground truth shortest path via an MSE loss. The input to the shortest path algorithm is a cost embedding matrix predicted by a neural network.

## B.4   Stochastic Smoothing

Another differentiation method is stochastic smoothing [64]. This method regularizes a non-differentiable and discontinuous loss function $\ell(y)$ by randomly perturbing its input with random noise $\epsilon$ (i.e., $\ell(y + \epsilon)$). The loss function is then approximated as $\ell(y) \approx \ell_\epsilon(y) = \mathbb{E}_\epsilon[\ell(y + \epsilon)]$. While $\ell$ may be non-differentiable, its smoothed stochastic counterpart $\ell_\epsilon$ is differentiable and the corresponding gradient and Hessian can be estimated via the following result.

**Lemma 1** (Exponential Family Smoothing, adapted from Lemma 1.5 in Abernethy *et al.* [64]). *Given a distribution over $\mathbb{R}^m$ with a probability density function $\mu$ of the form $\mu(\epsilon) = \exp(-\nu(\epsilon))$ for any twice-differentiable $\nu$, then*

$$\nabla_y l_\epsilon(y) \;\;=\;\; \nabla_y \mathbb{E}_\epsilon\left[\ell(y + \epsilon)\right] \;\;=\;\; \mathbb{E}_\epsilon\left[\ell(y + \epsilon)\,\nabla_\epsilon \nu(\epsilon)\right], \tag{13}$$

$$\nabla_y^2 l_\epsilon(y) \;\;=\;\; \nabla_y^2 \mathbb{E}_\epsilon\left[\ell(y + \epsilon)\right] \;\;=\;\; \mathbb{E}_\epsilon\left[\ell(y + \epsilon)\left(\nabla_\epsilon \nu(\epsilon)\nabla_\epsilon \nu(\epsilon)^\top - \nabla_\epsilon^2 \nu(\epsilon)\right)\right]. \tag{14}$$

A *variance-reduced form* of (13) and (14) is

$$\nabla_y \mathbb{E}_\epsilon\left[\ell(y + \epsilon)\right] \;\;=\;\; \mathbb{E}_\epsilon\left[(\ell(y + \epsilon) - \ell(y))\,\nabla_\epsilon \nu(\epsilon)\right], \tag{15}$$

$$\nabla_y^2 \mathbb{E}_\epsilon\left[\ell(y + \epsilon)\right] \;\;=\;\; \mathbb{E}_\epsilon\left[(\ell(y + \epsilon) - \ell(y))\left(\nabla_\epsilon \nu(\epsilon)\nabla_\epsilon \nu(\epsilon)^\top - \nabla_\epsilon^2 \nu(\epsilon)\right)\right]. \tag{16}$$

In this work, we use this to estimate the gradient of the shortest path algorithm. By including the second derivative, we extend the perturbed optimizer losses to Newton losses. This also lends itself to full second-order optimization.

### B.4.1 SS of Algorithm

SS of algorithm is an extension of this formulation, where stochastic smoothing is used to compute the Jacobian of the smoothed algorithm, e.g., $f : \mathbb{R}^{144} \to \mathbb{R}^{144}$ and the loss is, e.g., $\ell(y) = \text{MSE}(f(y), \text{label})$. Here, we can backpropagate through MSE and can apply stochastic smoothing to $f$ only. (The idea being that for many samples, it is better to estimate the Jacobian rather than the gradient of smoothing the entire loss.) While computing the Jacobian of $f$ is simple with stochastic smoothing, the Hessian here would be of size $144 \times 144 \times 144 \times 144$, making it infeasible to estimate this Hessian via sampling.

### B.5 Perturbed Optimizers with Fenchel-Young Losses

Berthet *et al.* [21] build on stochastic smoothing and Fenchel-Young losses [65] to propose perturbed optimizers with Fenchel-Young losses. For this, they use algorithms, like Dijkstra, to solve optimization problems of the type $\max_{w \in \mathcal{C}} \langle y, w \rangle$, where $\mathcal{C}$ denotes the feasible set, e.g., the set of valid paths. Berthet *et al.* [21] identify the argmax to be the differential of max, which allows a simplification of stochastic smoothing. By identifying similarities to Fenchel-Young losses, they find that the gradient of their loss is

$$\nabla_y \ell(y) = \mathbb{E}_\epsilon \left[ \arg\max_{w \in \mathcal{C}} \langle y + \epsilon, w \rangle \right] - w^\star \tag{17}$$

where $w^\star$ is the ground truth solution of the optimization problem (e.g., shortest path). This formulation allows optimizing the model without the need for computing the actual value of the loss function. Berthet *et al.* [21] find that the number of samples—surprisingly—only has a small impact on performance, such that 3 samples were sufficient in many experiments, and in some cases even a single sample was sufficient. In this work, we confirm this behavior and also compare it to plain stochastic smoothing. We find that for perturbed optimizers, the number of samples barely impacts performance, while for stochastic smoothing more samples always improve performance. If only few samples can be afforded (like 10 or less), perturbed optimizers are better as they are more sample efficient; however, when more samples are available, stochastic smoothing is superior as it can utilize more samples better.

## C Hyperparameters and Training Details

**Sorting and ranking.** 100,000 training steps with Adam and learning rate 0.001. Same convolutional network as in all prior works on the benchmark: Two convolutional layers with a kernel size of 5x5, 32 and 64 channels respectively, each followed by a ReLU and MaxPool layer; after flattening, this is followed by a fully connected layer with a size of 64, a ReLU layer, and a fully connected output layer mapping to a scalar.

- NeuralSort
  - Temperature $\tau = 1.0$ [Best for baseline from grid 0.001, 0.01, 0.1, 1, 10]
- SoftSort
  - Temperature $\tau = 0.1$ [Best for baseline from grid 0.001, 0.01, 0.1, 1, 10]
- Logistic DSN
  - Type: `odd_even`
  - Inverse temperature
    * For $n = 5$: $\beta = 10$ [Best for baseline from grid 10, 15, 20]
    * For $n = 10$: $\beta = 10$ [Best for baseline from grid 10, 20, 40]
- Cauchy DSN
  - Type: `odd_even`
  - Inverse temperature
    * For $n = 5$: $\beta = 10$ [Best for baseline from grid 10, 100]
    * For $n = 10$: $\beta = 100$ [Best for baseline from grid 10, 100]

**Shortest-path.** Model, Optimizer, LR schedule, and epochs same as in prior work: First block of ResNet18, Adam optimizer, training duration of 50 epochs, and a learning rate decay by a factor of 10 after 30 and 40 epochs each.

- AlgoVision:
    - $\beta = 10$
    - n_iter=18
      (number iterations in for loop / max num iteration in algovision while loop)
    - t_conorm='probabilistic'
    - Initial learning rate, for each (as it varies between for/while loop and L1/L2 loss), best among [1, 0.33, 0.1, 0.033, 0.01, 0.0033, 0.001].
- SS of loss / algorithm:
    - Distribution: Gaussian with $\sigma = 0.1$
      [best $\sigma$ on factor 10 exponential grid for baseline]
    - Initial LR: 0.001
- PO / FY loss:
    - Distribution: Gaussian with $\sigma = 0.1$
      [best $\sigma$ on factor 10 exponential grid for baseline]
    - Initial LR: 0.01

### C.1 Hyperparameter $\lambda$

For the experiments in the tables, select $\lambda$ based one seed from the grid $\lambda \in [0.001, 0.01, 0.1, 1, 10, 100, 100, 1000, 3000]$. For the experiments in Tables 2 and 5, we present the values in the Tables 4 and 5, respectively. For the experiments in Table 2, we use a Tikhonov regularization strength of $\lambda = 1000$ for the $L_1$ variants and $\lambda = 3000$ for the $L_2^2$ variants.

Table 4: Tikhonov regularization strengths $\lambda$ for the experiment in Table 1.

|  | $n = 5$ | | | | $n = 10$ | | | |
| --- | --- | --- | --- | --- | --- | --- | --- | --- |
|  | NeuralSort | SoftSort | Logistic DSN | Cauchy DSN | NeuralSort | SoftSort | Logistic DSN | Cauchy DSN |
| NL (Hessian) | $\lambda = 0.01$ | $\lambda = 10$ | $\lambda = 0.1$ | $\lambda = 0.1$ | $\lambda = 0.01$ | $\lambda = 1$ | $\lambda = 0.1$ | $\lambda = 0.1$ |
| NL (Fisher) | $\lambda = 0.1$ | $\lambda = 10$ | $\lambda = 0.1$ | $\lambda = 0.1$ | $\lambda = 100$ | $\lambda = 100$ | $\lambda = 0.1$ | $\lambda = 0.1$ |

Table 5: Tikhonov regularization strengths $\lambda$ for the experiment in Table 3.

| Loss | SS of loss | | | SS of algorithm | | | PO w/ FY loss | | |
| --- | --- | --- | --- | --- | --- | --- | --- | --- | --- |
| # Samples | 3 | 10 | 30 | 3 | 10 | 30 | 3 | 10 | 30 |
| NL (Hessian) | $\lambda = 1000$ | $\lambda = 1000$ | $\lambda = 1000$ | — | — | — | $\lambda = 1000$ | $\lambda = 1000$ | $\lambda = 1000$ |
| NL (Fisher) | $\lambda = 0.1$ | $\lambda = 0.1$ | $\lambda = 0.1$ | $\lambda = 1000$ | $\lambda = 1000$ | $\lambda = 1000$ | $\lambda = 1000$ | $\lambda = 1000$ | $\lambda = 1000$ |

### C.2 List of Assets

- Multi-digit MNIST [37], which builds on MNIST [70]   [MIT License / CC License]
- Warcraft shortest-path data set [22]   [MIT License]
- PyTorch [71]   [BSD 3-Clause License]

## D   Runtimes

In this supplementary material, we provide and discuss runtimes for the experiments. All times are of full training on a single A6000 GPU.

In the differentiable sorting and ranking experiment, as shown in Table 6, we observe that the runtime from regular training compared to the Newton loss with the Fisher is only marginally increased. This is because computing the Fisher and inverting it is very inexpensive. We observe that the Newton loss

with the Hessian, however, is more expensive: due to the implementation of the differentiable sorting and ranking operators, we compute the Hessian by differentiating each element of the gradient, which makes this process fairly expensive. An improved implementation could make this process much faster. Nevertheless, there is always some overhead to computing the Hessian compared to the Fisher.

Table 6: Runtimes [h:mm] for the differentiable sorting results corresponding to Table 1.

|  | $n = 5$ | | | $n = 10$ | | |
| --- | --- | --- | --- | --- | --- | --- |
|  | DSN | NeuralSort | SoftSort | DSN | NeuralSort | SoftSort |
| Baseline | 1:10 | 1:02 | 1:01 | 1:43 | 1:27 | 1:24 |
| NL (Hessian) | 2:24 | 1:07 | 1:10 | 6:17 | 1:42 | 1:40 |
| NL (Fisher) | 1:11 | 1:03 | 1:02 | 1:44 | 1:27 | 1:25 |

In Table 7, we show the runtimes for the shortest-path experiment with AlgoVision. Here, we observe that the runtime overhead is very small.

Table 7: Runtimes [h:mm] for the shortest-path results corresponding to Table 2.

| Algorithm Loop | For | | While | |
| --- | --- | --- | --- | --- |
| Loss | $L_1$ | $L_2^2$ | $L_1$ | $L_2^2$ |
| Baseline | 0:10 | 0:10 | 0:15 | 0:15 |
| NL (Fisher) | 0:10 | 0:11 | 0:15 | 0:15 |

In Table 8, we show the runtimes for the shortest-path experiment with stochastic methods. Here, we observe that the runtime overhead is also very small. Here, the Hessian is also cheap to compute as it is not computed with automatic differentiation.

Table 8: Runtimes [h:mm] for the shortest-path results corresponding to Table 3.

| Loss | SS of loss | | | SS of algorithm | | | PO w/ FY loss | | |
| --- | --- | --- | --- | --- | --- | --- | --- | --- | --- |
| # Samples | 3 | 10 | 30 | 3 | 10 | 30 | 3 | 10 | 30 |
| Baseline | 0:15 | 0:23 | 0:53 | 0:15 | 0:23 | 0:53 | 0:11 | 0:19 | 0:49 |
| NL (Hessian) | 0:15 | 0:23 | 0:53 | – | – | – | 0:11 | 0:19 | 0:50 |
| NL (Fisher) | 0:15 | 0:23 | 0:54 | 0:15 | 0:23 | 0:53 | 0:11 | 0:19 | 0:50 |

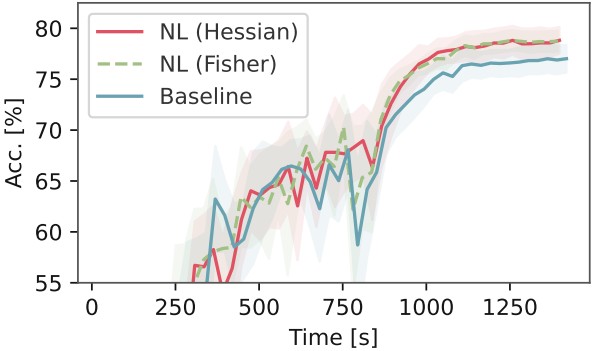

Figure 5: Training time plot corresponding to Figure 3: Test accuracy (perfect matches) plot for 'SS of loss' with 10 samples on the Warcraft shortest-path benchmark.

# E  Equivalences under the Split

Using gradient descent step according to (1) is equivalent to using two gradient steps of the alternating scheme (2), namely one step for (2a) and one step for (2b). This has also been considered by [62] in a different context.

**Lemma 2** (Gradient Descent Step Equality between (1) and (2a)+(2b)). *A gradient descent step according to* (1) *with arbitrary step size $\eta$ coincides with two gradient descent steps, one according to* (2a) *and one according to* (2b)*, where the optimization over $\theta$ has a step size of $\eta$ and the optimization over $z$ has a unit step size.*

*Proof.* Let $\theta \in \Theta$ be the current parameter vector and let $\mathbf{z} = f(\mathbf{x}; \theta)$. Then the gradient descent steps according to (2a) and (2b) with step sizes 1 and $\eta > 0$ are expressed as

$$\mathbf{z} \leftarrow \mathbf{z} - \nabla_{\mathbf{z}}\,\ell(\mathbf{z}) = f(\mathbf{x}; \theta) - \nabla_f\,\ell(f(\mathbf{x}; \theta)) \tag{18}$$

$$\theta \leftarrow \theta - \eta\,\nabla_\theta\,\tfrac{1}{2}\|\mathbf{z} - f(\mathbf{x}; \theta)\|_2^2$$

$$= \theta - \eta\,\frac{\partial\,f(\mathbf{x}; \theta)}{\partial\,\theta} \cdot (f(\mathbf{x}; \theta) - \mathbf{z})\,. \tag{19}$$

Combining (18) and (19) leads to

$$\theta \leftarrow \theta - \eta\,\frac{\partial\,f(\mathbf{x}; \theta)}{\partial\,\theta} \cdot (f(\mathbf{x}; \theta) - f(\mathbf{x}; \theta) + \nabla_f\,\ell(f(\mathbf{x}; \theta)))$$

$$= \theta - \eta\,\nabla_\theta\,\ell(f(\mathbf{x}; \theta)), \tag{20}$$

which is exactly a gradient descent step starting at $\theta \in \Theta$ with step size $\eta$. $\qquad\square$

Moreover, we show that a corresponding equality also holds for a special case of the Newton step.

**Lemma 3** (Newton Step Equality between (1) and (2a)+(2b) for $m = 1$)**.** *In the case of $m = 1$ (i.e., a one-dimensional output), a Newton step according to (1) with arbitrary step size $\eta$ coincides with two Newton steps, one according to (2a) and one according to (2b), where the optimization over $\theta$ has a step size of $\eta$ and the optimization over $z$ has a unit step size.*

*Proof.* Let $\theta \in \Theta$ be the current parameter vector and let $\mathbf{z} = f(\mathbf{x}; \theta)$. Then applying Newton steps according to (2a) and (2b) leads to

$$\mathbf{z} \leftarrow \mathbf{z} - (\nabla_{\mathbf{z}}^2 \ell(\mathbf{z}))^{-1} \nabla_{\mathbf{z}}\,\ell(\mathbf{z})$$

$$= f(\mathbf{x}; \theta) - (\nabla_f^2 \ell(f(\mathbf{x}; \theta)))^{-1} \nabla_f \ell(f(\mathbf{x}; \theta)) \tag{21}$$

$$\theta \leftarrow \theta - \eta\left(\nabla_\theta^2 \frac{1}{2}\|\mathbf{z} - f(\mathbf{x}; \theta)\|_2^2\right)^{-1} \nabla_\theta \frac{1}{2}\|\mathbf{z} - f(\mathbf{x}; \theta)\|_2^2 \tag{22}$$

$$= \theta - \eta\left(\frac{\partial}{\partial\theta}\left[\frac{\partial\,f(\mathbf{x}; \theta)}{\partial\,\theta} \cdot (f(\mathbf{x}; \theta) - \mathbf{z})\right]\right)^{-1} \frac{\partial\,f(\mathbf{x}; \theta)}{\partial\,\theta} \cdot (f(\mathbf{x}; \theta) - \mathbf{z}) \tag{23}$$

$$= \theta - \eta\left(\frac{\partial}{\partial\theta}\left[\frac{\partial\,f(\mathbf{x}; \theta)}{\partial\,\theta}\right](f(\mathbf{x}; \theta) - \mathbf{z}) + \left(\frac{\partial\,f(\mathbf{x}; \theta)}{\partial\,\theta}\right)^2\right)^{-1} \frac{\partial\,f(\mathbf{x}; \theta)}{\partial\,\theta} \cdot (f(\mathbf{x}; \theta) - \mathbf{z})$$

Inserting (21), we can rephrase the update above as

$$\theta \leftarrow \theta - \eta\left(\frac{\partial}{\partial\theta}\left[\frac{\partial\,f(\mathbf{x}; \theta)}{\partial\,\theta}\right](\nabla_f^2\ell(f(\mathbf{x}; \theta)))^{-1}\nabla_f\ell(f(\mathbf{x}; \theta)) + \left(\frac{\partial\,f(\mathbf{x}; \theta)}{\partial\,\theta}\right)^2\right)^{-1}$$

$$\cdot \frac{\partial\,f(\mathbf{x}; \theta)}{\partial\,\theta} \cdot (\nabla_f^2\ell(f(\mathbf{x}; \theta)))^{-1}\nabla_f\ell(f(\mathbf{x}; \theta)) \tag{24}$$

By applying the chain rule twice, we further obtain

$$\nabla_\theta^2\ell(f(\mathbf{x}; \theta)) = \frac{\partial}{\partial\theta}\left[\frac{\partial\,f(\mathbf{x}; \theta)}{\partial\,\theta}\nabla_f\ell(f(\mathbf{x}; \theta))\right]$$

$$= \frac{\partial}{\partial\theta}\left[\frac{\partial\,f(\mathbf{x}; \theta)}{\partial\,\theta}\right]\nabla_f\ell(f(\mathbf{x}; \theta)) + \frac{\partial\,f(\mathbf{x}; \theta)}{\partial\,\theta}\frac{\partial}{\partial\theta}\nabla_f\ell(f(\mathbf{x}; \theta))$$

$$= \frac{\partial}{\partial\theta}\left[\frac{\partial\,f(\mathbf{x}; \theta)}{\partial\,\theta}\right]\nabla_f\ell(f(\mathbf{x}; \theta)) + \frac{\partial\,f(\mathbf{x}; \theta)}{\partial\,\theta}\nabla_f\frac{\partial}{\partial\theta}\ell(f(\mathbf{x}; \theta))$$

$$= \frac{\partial}{\partial\theta}\left[\frac{\partial\,f(\mathbf{x}; \theta)}{\partial\,\theta}\right]\nabla_f\ell(f(\mathbf{x}; \theta)) + \left(\frac{\partial\,f(\mathbf{x}; \theta)}{\partial\,\theta}\right)^2\nabla_f^2\ell(f(\mathbf{x}; \theta)),$$

which allows us to rewrite (24) as

$$\theta' = \theta - \left((\nabla_f^2\ell(f(\mathbf{x}; \theta)))^{-1}\nabla_\theta^2\ell(f(\mathbf{x}; \theta))\right)^{-1}(\nabla_f^2\ell(f(\mathbf{x}; \theta)))^{-1}\nabla_\theta\,\ell(f(\mathbf{x}; \theta))$$

$$= \theta - (\nabla_\theta^2\ell(f(\mathbf{x}; \theta)))^{-1}\nabla_\theta\,\ell(f(\mathbf{x}; \theta)),$$

which is exactly a single Newton step starting at $\theta \in \Theta$. $\qquad\square$

# F  Woodbury Matrix Identity for High Dimensional Outputs

For the empirical Fisher method, where $\mathbf{z}^\star \in \mathbb{R}^{N \times m}$ is computed via

$$z_i^\star = \bar{y}_i - \left( \tfrac{1}{N} \sum_{j=1}^N \nabla_{\bar{y}_j} \ell(\bar{\mathbf{y}}) \, \nabla_{\bar{y}_j} \ell(\bar{\mathbf{y}})^\top + \lambda \cdot \mathbf{I} \right)^{-1} \nabla_{\bar{y}_i} \ell(\bar{\mathbf{y}})$$

for all $i \in \{1, ..., N\}$ and $\bar{\mathbf{y}} = f(\mathbf{x}; \theta)$, we can simplify the computation via the Woodbury matrix identity [66]. In particular, we can simplify the computation of the inverse to

$$\left( \tfrac{1}{N} \sum_{j=1}^N \nabla_{\bar{y}_j} \ell(\bar{\mathbf{y}}) \, \nabla_{\bar{y}_j} \ell(\bar{\mathbf{y}})^\top + \lambda \cdot \mathbf{I} \right)^{-1} \tag{25}$$

$$= \left( \tfrac{1}{N} \nabla_{\bar{\mathbf{y}}} \ell(\bar{\mathbf{y}})^\top \nabla_{\bar{\mathbf{y}}} \ell(\bar{\mathbf{y}}) + \lambda \cdot \mathbf{I} \right)^{-1} \tag{26}$$

$$= \tfrac{1}{\lambda} \cdot \mathbf{I}_m - \tfrac{1}{N \cdot \lambda^2} \cdot \nabla_{\bar{\mathbf{y}}} \ell(\bar{\mathbf{y}})^\top \cdot \left( \tfrac{1}{N \cdot \lambda} \nabla_{\bar{\mathbf{y}}} \ell(\bar{\mathbf{y}}) \, \nabla_{\bar{\mathbf{y}}} \ell(\bar{\mathbf{y}})^\top + \mathbf{I}_N \right)^{-1} \cdot \nabla_{\bar{\mathbf{y}}} \ell(\bar{\mathbf{y}}) \,. \tag{27}$$

This reduces the cost of matrix inversion from $\mathcal{O}(m^3)$ down to $\mathcal{O}(N^3)$. We remark that this variant is only helpful for cases where the batch size $N$ is smaller than the output dimensionality $m$.

Further, we remark that in a case where the output dimensionality $m$ and the batch size $N$ are both very large, i.e., the cost of the inverse is typically still small in comparison to the computational cost of the neural network. To illustrate this, the dimensionality of the last hidden space (i.e., before the last layer) $h_l$ is typically larger than the output dimensionality $m$. Accordingly, the cost of only the last layer itself is $\mathcal{O}(N \cdot m \cdot h_l)$, which is typically larger than the cost of the inversion in Newton Losses. In particular, assuming $h_l > m$, $\mathcal{O}(N \cdot m \cdot h_l) > \mathcal{O}(\min(m, N)^3)$.

We remark that the Woodbury inverted variation can also be used in `InjectFisher`.

# G  InjectFisher for CVXPY Layers

In this section, we provide an extension of the presented method, from algorithmic losses to optimizing the parameters in differentiable convex optimization layers [72]. For this, we utilize the `cvxpylayers` framework, and, as a proof of concept apply it to the training of a 3-layer network, where the first 2 layers are modelled via `cvxpylayers`: https://github.com/cvxgrp/cvxpylayers/blob/master/examples/torch/ReLU%20Layers.ipynb.

Because there are no Hessians or second-order derivatives available in `cvxpylayers`, we applied the empirical Fisher variant, via the `InjectFisher` function, to `cvxpylayers`. We apply `InjectFisher` to (the vector of) all weights and biases of a given layer, and choose $\lambda = 0.1$.

We ran it for 5 seeds, and the loss improves in each case (see Table 9). Importantly, we note that these are paired results (only networks with the same initialization are comparable).

Table 9: Loss of a `cvxpylayers` model. Results comparable only within each seed.

| Seed | cvxpy | cvxpy + InjectFisher |
|------|-------|----------------------|
| 1 | 0.322 | **0.185** |
| 2 | 0.186 | **0.157** |
| 3 | 0.522 | **0.247** |
| 4 | 0.225 | **0.179** |
| 5 | 0.260 | **0.211** |

# H  Gradient Visualizations

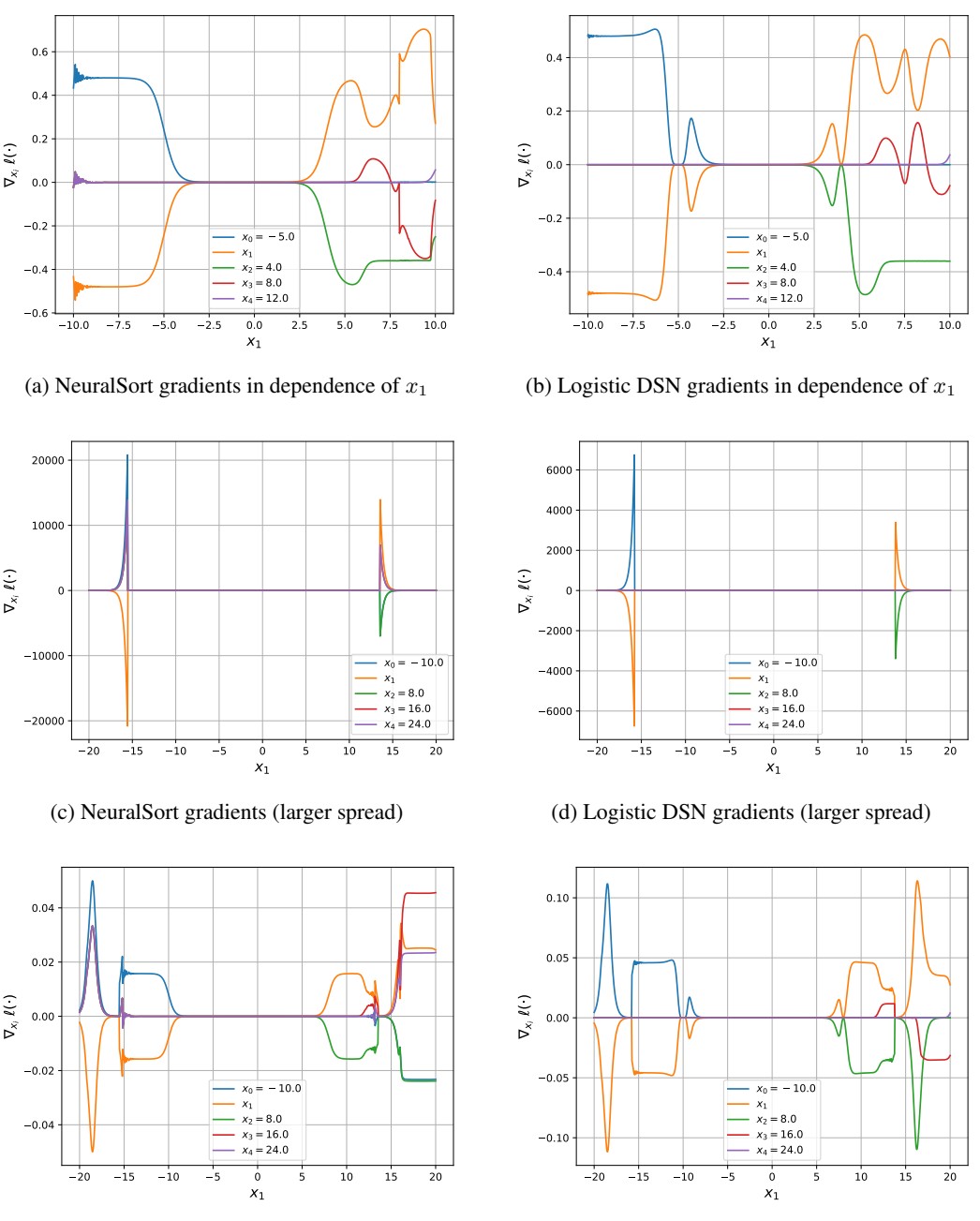

(a) NeuralSort gradients in dependence of $x_1$

(b) Logistic DSN gradients in dependence of $x_1$

(c) NeuralSort gradients (larger spread)

(d) Logistic DSN gradients (larger spread)

(e) Newton loss gradients (empirical Fisher) for NeuralSort (larger spread)

(f) Newton loss gradients (empirical Fisher) for Logistic DSN (larger spread)

Figure 6: We illustrate the gradient of the NeuralSort and logistic DSN losses in dependence of one of the five input dimensions for the $n = 5$ case. In the illustrated example, one can observe that both algorithms experience exploding gradients when the inputs are too far away from each other (which is also controllable via steepness $\beta$ / temperature $\tau$), see Figures (c) / (d). Further, we can observe that the gradients themselves can be quite chaotic, making the optimization of the loss rather challenging. In Figures (e) / (f), we illustrate how using the empirical Fisher Newton Loss recovers from the exploding gradients experienced in (c) / (d). The input examples are a simplification of actual inputs, as here, $x_0, x_2, x_3, x_4$ are already in their correct order, and having multiple disagreements makes the loss more chaotic.

